# MetaAgent: Automatically Building Multi-Agent System based on Finite State Machine

## Abstract

Large Language Models (LLMs) can solve various practical tasks via a multi-agent system. However, existing human-designed multi-agent systems can only adapt to a limited number of pre-defined scenarios. Current auto-designed methods also have several drawbacks, including no tool support, reliance on external data, and inflexible communication structure. Therefore, we propose **MetaAgent**, a novel framework to automatically generate a multi-agent system based on a finite state machine. Given a task description, MetaAgent will design a multi-agent system and polish it through self-generated test queries. When the multi-agent system is deployed, the finite state machine, which supports the traceback and is more suitable for tool-using, will control the process of problem-solving. To evaluate our framework, we conduct experiments on both practical tasks and basic NLP tasks, the results indicate that the generated multi-agent system surpasses other auto-designed methods and can achieve a comparable performance with the human-designed multi-agent system which is polished for those specific tasks.

## 1 Introduction

Large Language Models (LLMs) (OpenAI et al. (2024); Zhao et al. (2024)) show a spring-up of intelligence, containing strong ability of coding, reasoning, and numerous compressed knowledge. Utilizing LLM as the brain to build agents can complement various complex tasks, which requires the agent to plan, utilize tools, and make reflections. (Yao et al. (2023); Shinn et al. (2023); Wang et al. (2024a); Qin et al. (2023)). To further improve the performance, the multi-agent system has proposed, which improves and enlarges the abilities of the agent by assigning different roles and skills to LLMs and designing effective cooperation mechanisms to organize them (Hong et al. (2023); Qian et al. (2024); Yan et al. (2024); Huang et al. (2024)). Despite the success, most of the existing multi-agent are still manually designed, introducing human efforts to implement the complex codebase and needing several iterations of human polishing. Moreover, these frameworks are built only to solve tasks in some specific scenarios, further enhancing the design cost.

To address it, a few works try to build multi-agent systems automaticallyChen et al. (2024a); Wang et al. (2024d); Yuan et al. (2024). However, current works have failed to construct a complete and practical multi-agent system due to several reasons. SPP, AutoAgents, and EvoAgent (Chen et al. (2024a); Wang et al. (2024d); Yuan et al. (2024)) design multi-agent systems for each specific case. In other words, the produced multi-agent system can only handle the specific case and **lacks generalization** to other cases in the same task domain. Some of them **do not support tool-using** as well. ADAS and Symbolic-Learning (Hu et al. (2024); Zhou et al. (2024)) build multi-agent systems automatically based on self-iteration algorithms. However, tons of iterations and **external data are needed** for optimization. Moreover, following the communication structure of human-designed multi-agent systems (Hong et al. (2023); Qian et al. (2024); Du et al. (2023)), current works use a linear cooperation structure to organize agents, simulating Standardized Operating Procedures (SOPs) in human society, which **can not trace back** to previous steps when encountering errors or misunderstanding.

To address the limitations of human-designed multi-agent systems and drawbacks of existing auto-design methods, we introduce **MetaAgent**: A framework that can automatically design **finite state machine** based multi-agent system for various types of tasks.

| Framework | MetaGPT | AutoAgents | SPP | EvoAgent | ADAS | Symbolic | MetaAgent |
|---|---|---|---|---|---|---|---|
| **Auto-Designed** | ✗ | ✓ | ✓ | ✓ | ✓ | ✓ | ✓ |
| **Generalization** | ✓ | ✗ | ✗ | ✗ | ✓ | ✓ | ✓ |
| **Tool Enabled** | ✗ | ✓ | ✗ | ✓ | ✗ | ✓ | ✓ |
| **Traceback Ability** | ✗ | ✗ | ✗ | ✗ | ✗ | ✗ | ✓ |
| **Non-External Data Depend** | ✓ | ✓ | ✓ | ✓ | ✗ | ✗ | ✓ |

Table 1: Comparison of existing and proposed Multi-Agent Frameworks

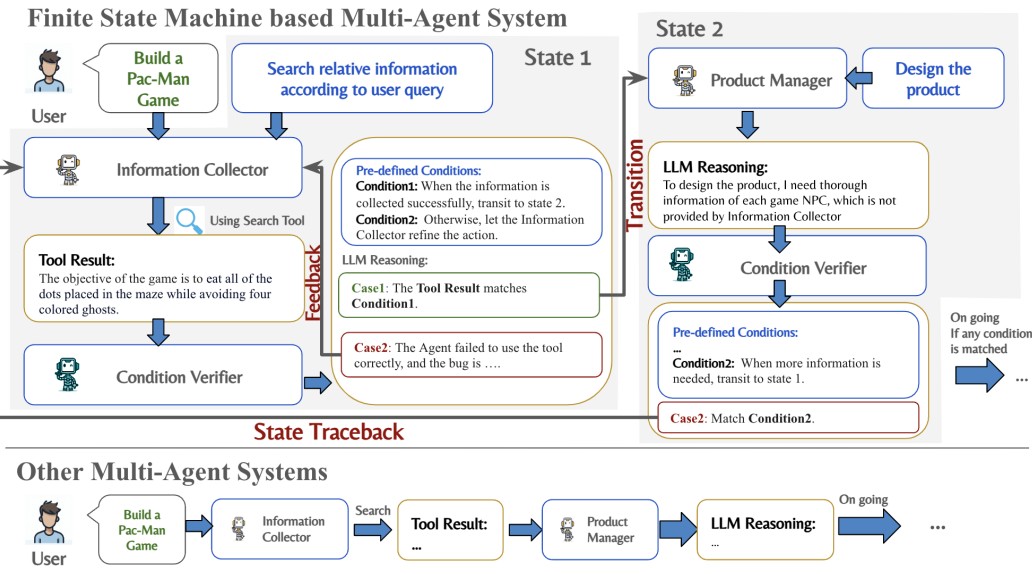

Figure 1: The above part shows an example of what is a state, and how our finite state machine structure works. The blow part shows how other linear-structured Multi-Agent Systems work.

Specifically, given a general description of a type of task, the MetaAgent will first design several agents needed to solve the task. Then, to organize these agents, several states are summarized based on the possible steps involved in solving the task. Each state includes the corresponding task-solving agent, the instructions for the task-solving agent, the condition verifier who checks whether the output meets certain state transition conditions, and the listener agents who will receive the output of the state. This design leverages the LLM's decision-making ability to dynamically manage the problem-solving process when encountering different cases within the given type of task.

The definition of state is inherently suitable for tool usage because it supports a multi-turn and dynamic environment. The condition verifier checks whether the previous action needs refinement or is complete to proceed to the next state. If errors occur during the tool-using process, the task-solving agent can refine its actions over several turns, enhancing robustness. Similarly, the condition verifier can trace the state back to the previous one if it detects errors or misunderstandings, ensuring a flexible workflow within the finite state machine. This machine acts as a guideline for problem-solving. In specific cases, the agent follows state instructions to generate state-by-state outcomes until reaching the final stage, where it submits the solutions to the user.

Before deploying the finite state machine based multi-agent system to solve practical tasks, we design a self-iteration mechanism to refine the system. A test generator is tasked with writing both primary and edge cases based on the tasks and initial design. The failure trajectories of these generated tests are analyzed by an adaptor, and the finite state machine is revised. Unlike relative works (Hu et al. (2024); Zhou et al. (2024)), the iteration method does not need external data as

well as numerous training steps. That's because the self-generated test which mainly helps optimize the FSM structure to avoid trivial states and long chains, is enough to ensure robust performance without needing carefully designed tests from external data or benchmarks.

When deployed, the multi-agent system can efficiently handle most cases within the task domain due to the finite state machine mechanism and prior testing on primary and edge cases. The user query, combined with the current state's instructions, serves as the input for the task-solving agent. The agent's output is sent to the state's condition verifier, which has several pre-defined state transition conditions in its system prompt. If a condition is met, the current state transitions to the corresponding state, which can also be a previous state, enabling the finite state machine's state traceback capability. Before the transition, the task-solving agent's output is sent to listeners as memory. Figure 1 illustrates the working mechanism of the finite state machine and compares it with other multi-agent systems with linear structures.

To verify that our MetaAgent is a general and robust framework capable of automatically producing customized multi-agent systems for various scenarios, we conduct experiments on realistic tasks. These include Machine Learning Bench (Hong et al. (2024)), software development tasks (Zhou et al. (2024)), and NLP tasks like Trivial Creative Writing (Wang et al. (2024d)), which are widely used to evaluate other auto-design multi-agent systems. The experiments indicate that the multi-agent system produced by the MetaAgent framework surpasses other automatic systems and achieves performance levels comparable to manually designed systems tailored for the tasks.In the Machine Learning tasks, the multi-agent system generated by MetaAgent achieved 97% of the average performance of the best human-designed multi-agent system, surpassing all other human-designed and multi-designed frameworks. In the software development task, MetaAgent passed 50% more checkpoints than the human-designed system. Our ablation study on tool usage, iteration, and traceback shows a 10% to 50% decrease in performance on the aforementioned tasks, highlighting the critical importance of these features.

## 2 RELATED WORKS

### 2.1 MULTI-AGENT SYSTEM

Previous works have discussed multi-agent systems in various scenarios. One category of Multi-Agent System is designed to simulate real-world scenarios (Park et al. (2023); Xu et al. (2024); Hua et al. (2024)). Researchers can find some rules or conduct social experiments in these systems.

In this research, we focus on the multi-agent system which builds for problem-solving. Early works use merely the reasoning ability of LLM to build systems like debating, voting, and negotiating. (Wu et al. (2023); Du et al. (2023); Yan et al. (2024); Bianchi et al. (2024)) Later works implement tool-using and more complex communication structures for the system. MetaGPT and ChatDev (Qian et al. (2024); Hong et al. (2023)) build a Multi-Agent System for software development and introduce a message pool to manage communication. DataInterpreter and AgentCoder (Hong et al. (2024); Huang et al. (2024)) focus on data science or Python code problems but are also limited to pre-defined scenarios. There are a few works that apply the finite state machine to control the agentic system. (Wu et al. (2024); Liu et al. (2024); Chen et al. (2024b)) But they are limited to certain scenarios as well as using a fixed method to detect certain output strings as the transition function, which is hard to adapt to complex real-world scenarios.

As the growing trend of automatic design, SPP (Wang et al. (2024d)) introduces a prompt-based method to build a linear multi-agent system for each case of task, invoking the compressed knowledge by assigning the roles. AutoAgents (Chen et al. (2024a)) is built on the codebase of MetaGPT and further improves the Multi-Agent System by adapting planning and multi-turn cooperation between agents. ADAS and Symbolic Learning (Hu et al. (2024); Zhou et al. (2024)) try to optimize a multi-agent system from a given simple system, but they need many iterations and focus more on the inner structure of each single agent. However, there is a lack of a method to efficiently and automatically build a tool-enabled multi-agent system that can handle a specific domain.

## 2.2 Tool LLM

Utilizing tools is a significant feature of LLM Agent as well as our MetaAgent Framework, for it enables the Agents to interact with external worlds, enlarging their ability scope. Previous works on tool LLM can be divided into two categories. The first category (Patil et al. (2023); Qin et al. (2023)) teaches LLMs to utilize a wide range of real-world APIs via function-calling, with a focus on the breadth of tools. The second category focuses on the usage of some specific tools like search engines and code interpreters that can complete multiple tasks. CodeAct (Wang et al. (2024b)) first assigned code as actions and integrated various functions into the Python code snippet. PyBench and MINT (Zhang et al. (2024); Wang et al. (2024c)) evaluate LLM equipped with code interpreter on multiple tasks. Gao et al. (2024) shows LLM Agent equipped with a search engine has a significant ability growth in numerous information-seeking tasks. Our MetaAgent, mainly equipped the agents with code interpreter and search engine, promoting the tool-using ability to the area of automatic multi-agent system.

# 3 Method

## 3.1 Background

We first introduce the finite state machine to describe a multi-agent system. A finite state machine (FSM) is a computational model consisting of a finite number of states, and transition functions between those states (Hopcroft et al. (2001); Carroll & Long (1989)). In our setting, a state means one possible step when solving a problem, containing the task-solving agent, the condition verifier, the state instruction, and the listeners who receive the output when the state is complete. The state transition conditions are described by strings, which will be the basis for decision-making for the condition verifier. Hence, an FSM can be defined by a tetrad: $\{\Sigma, S, s_0, con\}$. The key concepts of a finite state machine consist of the following:

- $\Sigma$: The input string of the finite state machine.
- $S$: The set of states.
- $s_0$: The initial state, an element of $S$.
- $con$: State transition conditions.

The FSM will start at the initial state and transition between states under the control of state transition conditions until it either reaches the final state, indicating task completion or hits the maximum number of transitions, indicating task failure.

## 3.2 Construction Stage

**Agents Design**   Given the general descriptions of the task, the designer will first design several required agents that may be needed to solve the task. Each agent has the name, system prompt, and equipped tools selected from a pre-defined pool.

**Finite State Machine Design**   The designer generates a finite state machine based on the agents and task description. This finite state machine includes descriptions of each state and the conditions for state transitions. The design process involves several steps. Firstly, the designer should consider the various scenarios that may arise while solving different cases within the task domain. Based on these potential situations, several states that reflect these scenarios are created. For each state, the corresponding agent capable of addressing the situation is assigned, along with specific instructions for the agent. Next, the designer ensures that each state's output is received by the relevant agents by setting up listeners for each state. Finally, the states are connected by defining the conditions under which one state should transition to another.

**Test Case Generation**   After the first version of the multi-agent system is generated, the test generator designs several test queries based on the task description and the multi-agent system. To identify the drawbacks of the current system, the generator writes two types of queries. The first type covers the primary cases in the task domain, aiming to test the robustness of the current system. The second type consists of edge cases, which help the system become more complete.

**Construction Stage**

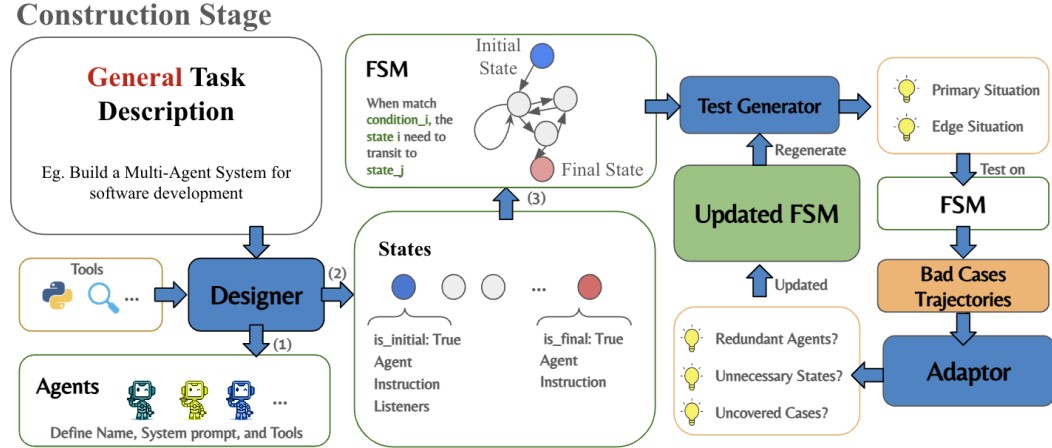

Figure 2: The construction stage of MetaAgent

**Self-Iteration**    By testing the multi-agent system on generated queries, we obtain the trajectories of bad cases. The adaptor is then prompted to update the multi-agent system from several aspects. First, identify any overlap in the agents' roles and determine if the agents can be combined. Next, detect any unnecessary states causing redundant information flow and simplify the states. Additionally, update the instructions or system prompts to handle edge cases. After these updates, the new multi-agent system is sent back to the test generator for targeted test queries. The multi-agent system can be fine-tuned after one or two iterations.

## 3.3 DEPLOYMENT STAGE

After the construction stage, the multi-agent system is fixed and ready for deployment in practical scenarios. In a specific task domain, the finite state machine operates according to Algorithm 1. Initially, the state is set to $s_0$, and the agent in this state acts based on the given instructions and query. The output, which is a combination of LLM text and tool responses (if used), is evaluated by the condition verifier using the system prompt containing the transition conditions. Given the output and conditions, the verifier assesses whether a condition is met and identifies the target state for transition. If a condition is met, the state transitions to the detected target state and the output of the current state is inserted into the memory of the listener agent, ensuring the flow of information. If the transition function indicates that the state is not complete for no condition is met, the finite state machine will continue to call the current agent until a transition condition is met or the maximum number of interactions $M$ is exceeded. Figure 1 shows an example of how a finite state machine works.

## 3.4 FEATURES OF METAAGENT

We discuss key features of MetaAgent that distinguish it from other human-design or auto-design multi-agent systems in this section.

**Suitable for Tool-Using**    In the area of utilizing LLM to solve complex and practical tasks, it is crucial to have the opportunity to refine or debug as well as call the tool for multi-turns to solve complex tasks that can not be solved in one turn. The structure of the finite state machine is naturally suitable for the above features because the condition verifier can continually urge the task-solving agent to debug or go a step further whenever the output does not match any state transition conditions.

**Enable State Traceback**    In the general problem-solving process, it is inevitable to encounter errors or misunderstandings from previous steps. Existing multi-agent systems with linear structures, such as SOPs, do not account for this, as they only support a predefined linear pipeline. To address

---

**Algorithm 1** Deployment Stage

---

**Require:** specific case $Q$, max iterations $M$, Finite State Machine $\{\Sigma, S, s_0, con\}$. A state $s$ contains the corresponding agent $s.Agent$, the instruction to the agent $s.Ins$, the listener agent who will receive the state output $s.Lis$ and the condition verifier for the state $s.Ver$

1: $s \leftarrow s_0$
2: $c \leftarrow 0$
3: **while** $c < M$ **do**
4:    $output \leftarrow s.Agent(s.Ins, Q)$
5:    $s_{target} \leftarrow s.Ver(output)$
6:    **if** $s_{target} = None$ **then**
7:       $output \leftarrow s.Agent(s.Ins, output)$
8:       $c \leftarrow c + 1$
9:    **else**
10:      $s \leftarrow s_{target}$
11:      $c \leftarrow c + 1$
12:      **for** Lis in s.Lis **do**
13:         $memory\_insert(Lis, output)$
14:      **end for**
15:   **end if**
16: **end while**

---

this weakness, our finite state machine enables state traceback. When the condition verifier identifies dilemmas caused by misunderstandings or failures in previous states, it transitions back to the previous state for refinement. For example, in a software development task, if the QA Test Agent finds that a file has not been written, it can trace back to the stage where the programmer writes the software to the file and provides debug information to the programmer.

**Interation by itself**    Compared to other works that depend on external and even in-bag data for training or optimization, MetaAgent can generate test queries on itself. We the initial version of FSM always failed because the designed agent and state are too trivial, which leads to an extremely long chain from the initial state to the final state. This also caused a large overlap in the work of many agents, which affected the efficiency of cooperation and task completion. Thus, the main purpose of iteration is to optimize the structure of FSM, ensuring it can work robustly. In other words, the self-generated test is enough for the iteration, and there is no need to carefully design tests from the external data or benchmarks.

**Handle Every Case in the Domain**    Figure 3 illustrates the various configurations of our MetaAgent compared to other Auto-Design Frameworks, including SPP, EvoAgent, and AutoAgents. Given a task domain, such as "A multi-agent system for software development" or "A multi-agent system for machine learning tasks," our MetaAgent designs a unified Multi-Agent System capable of addressing every case within the domain and generating corresponding solutions. In contrast, the other frameworks mentioned design distinct multi-agent systems for each specific case, which is less practical and more costly.

## 4 EXPERIMENT

We conduct a series of experiments on different tasks to show the versatility and robustness of our framework. We first compare MetaAgent on practical tasks including machine learning and software development tasks to show that the generated FSM-based multi-agent system surpasses other auto-design methods significantly and has comparable performance with a human-designed multi-agent system. After that, we also conducted experiments on Trivial Creative Writing, an NLP task requiring the Agent to gather knowledge in various domains, aiming to compare MetaAgent with other auto-design multi-agent systems. Ablation studies on tool-using, traceback, and iteration are

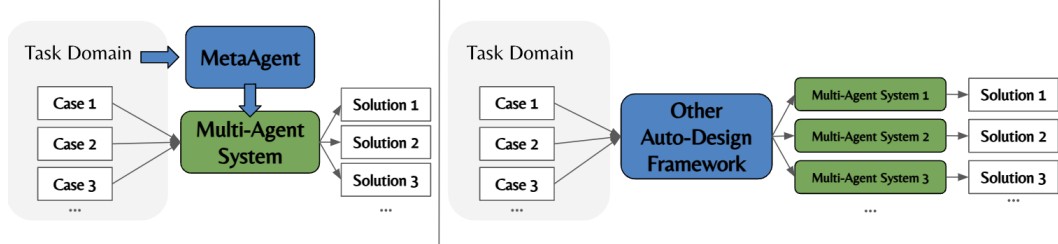

Figure 3: The difference between Task-Level Design (Left) and Case-Level Design (Right)

also conducted to reveal their impacts. We selected GPT-4o as the foundation model in experiments. The code interpreter and search engine are listed in the tool pool for selection.

## 4.1 REAL-WORLD CODING TASKS

### 4.1.1 MACHINE LEARNING BENCH

Machine Learning Bench(ml_bench) (Hong et al. (2024)) is a benchmark that requires agents to train a machine-learning model for regression or classification. We use the normalized performance score (NPS) as the metric to evaluate the quality of the trained machine learning model on the given evaluation datasets.

**Baselines** We select both human-designed and auto-designed Frameworks as baselines. AutoGen (Wu et al. (2023)), OpenIterpreter (Lucas (2023)), TaskWeaver (Qiao et al. (2024)), and DataInterpreter (Hong et al. (2024)) are typical human-designed multi-agent frameworks. We then adapt SPP (Wang et al. (2024d)) and AutoAgents (Chen et al. (2024a)) to the ml_bench by extracting the generated code and getting the execution result.

**Results and Analysis** Table 2 presents the results on ml_bench. The multi-agent system generated by MetaAgent outperforms all other auto-designed frameworks, which lack the mechanism to utilize tool feedback and thus process the dataset with hallucinations. MetaAgent also surpasses most human-designed multi-agent systems, demonstrating the robustness of its finite state machine. It achieves state-of-the-art (SOTA) performance on the Titanic and House Prices datasets and secures the second-highest scores on other datasets, showing comparable performance to DataInterpreter, a multi-agent system specifically tailored for machine learning tasks.

To analyze more deeply, we find that MetaAgent can generate a multi-agent system comprising a "Data Preparation and Model Selection Agent," a "Model Training Agent," and a "Report Agent." Following the designed state instructions, these agents can perform feature engineering, explore the dataset's structure, and pass the detected information to other agents. They can also train various models and report the best one. These features enable the multi-agent system to surpass others.

| Model / Task | Auto-Designed | Titanic | House Prices | SCTP | ICR | SVPC | Average |
|---|---|---|---|---|---|---|---|
| AutoGen | ✗ | 0.82 | 0.88 | 0.82 | 0.71 | 0.63 | 0.77 |
| Open Interpreter | ✗ | 0.81 | 0.87 | 0.52 | 0.25 | 0.00 | 0.49 |
| TaskWeaver | ✗ | 0.43 | 0.49 | 0.00 | 0.65 | 0.17 | 0.35 |
| Data Interpreter | ✗ | 0.82 | 0.91 | **0.89** | **0.91** | **0.77** | **0.86** |
| SPP | ✓ | 0.82 | 0.00 | 0.00 | 0.00 | 0.00 | 0.16 |
| AutoAgents | ✓ | 0.00 | 0.00 | 0.00 | 0.00 | 0.00 | 0.00 |
| MetaAgent | ✓ | **0.83** | **0.91** | 0.86 | 0.88 | 0.68 | 0.83 |

Table 2: Normalized performance score on ML Bench

### 4.1.2 SOFTWARE DEVELOPMENT

Software development is a comprehensive and practical task for evaluating agent systems, often used to assess various multi-agent frameworks. We have collected several representative software development tasks, including game and web app development. Unlike other software benchmarks (Zhou et al. (2024); Hong et al. (2023); Qian et al. (2024)), which primarily rely on subjective evaluation metrics, we have designed objective criteria for each software. These criteria include accessibility, functional completeness, and control ability (detailed in the Appendix). Each software is evaluated on four key points, earning one point for each test it passes. The metric used is the ratio of passed tests.

**Baselines** We select both human-designed and auto-designed multi-agent systems as baselines. MetaGPT (Hong et al. (2023)) designs a fixed SOP to organize the process of software development. We also adapt AutoAgents and SPP (Chen et al. (2024a); Wang et al. (2024d)) to the software development task by extracting the code they generated and save them to the files.

**Results and Analysis** Table 3 presents the results for five different software development tasks, demonstrating that our MetaAgent framework not only outperforms other auto-designed frameworks but also surpasses MetaGPT, a human-designed multi-agent framework for software development. Without tool-using capabilities, the performance of AutoAgents and SPP is significantly lower. Additionally, MetaGPT is constrained by its linear structure, which is lengthy and lacks the ability to trace back like a finite state machine.

The generated multi-agent system consists of a "Requirement Designer," a "Code Developer," and a "Tester." The tool-using and traceback features of the finite state machine contribute to its success. It can test whether the software can start and run smoothly via a code interpreter and trace back to the code development stage to fix bugs found in the testing state.

| Task / Model | MetaGPT | AutoAgents | SPP | MetaAgent |
|---|---|---|---|---|
| **Auto-Designed** | ✗ | ✓ | ✓ | ✓ |
| **2048 game** | 0.25 | 0 | 0.25 | **0.75** |
| **Snake game** | 0.25 | 0.75 | 0.50 | **1.0** |
| **Brick breaker game** | 0.75 | 0.25 | 0 | **0.50** |
| **Excel APP** | 0 | 0 | 0 | **1.0** |
| **Weather APP** | 0.50 | 0 | 0 | **1.0** |
| **Average** | 0.35 | 0.20 | 0.15 | **0.85** |

Table 3: Performance on Software Development Tasks

## 4.2 NLP TASK

### 4.2.1 TRIVIAL CREATIVE WRITING

Trivial Creative Writing is a demanding task that involves 100 instances. The model must craft a coherent narrative in this task while seamlessly integrating answers to N trivia questions. (Wang et al. (2024d)) The metric is the ratio of the number of trivia question keywords included in the story to the total number of trivia questions.

**Baselines** We select prompt engineering methods including Direct, CoT (Wei et al. (2023)), and Self-Refine (Madaan et al. (2023)) as well as auto-design methods like SPP, AutoAgents, and EvoAgent. (Wang et al. (2024d); Chen et al. (2024a); Yuan et al. (2024)) Note that, the selected auto-design methods all design multi-agent systems at the case level.

**Results and Analysis** The results of our experiments demonstrate three key findings. First, MetaAgent outperforms all other methods, achieving the highest score of 0.86 (Table 4). Second, methods incorporating tool-using capabilities show significant performance improvements, highlighting the importance of tool integration. Third, MetaAgent surpasses case-level multi-agent sys-

tems such as EvoAgent and AutoAgents, which score 0.84 and 0.82 respectively, demonstrating that case-level design is not only less unnecessary but also obviously more costly.

| Model / Task | Auto-Designed | Tool-Using | Case-Level Design | Score |
|---|---|---|---|---|
| **Direct** | ✗ | ✗ | ✗ | 0.75 |
| **CoT** | ✗ | ✗ | ✗ | 0.74 |
| **Self-Refine** | ✗ | ✗ | ✗ | 0.75 |
| **SPP** | ✓ | ✗ | ✓ | 0.79 |
| **AutoAgents** | ✓ | ✓ | ✓ | 0.82 |
| **EvoAgent** | ✓ | ✓ | ✓ | 0.84 |
| **MetaAgent** | ✓ | ✓ | ✗ | **0.86** |

Table 4: Trivial Creative Writing Performance

## 4.3 ABLATION STUDY

To demonstrate the importance of the key features of MetaAgent, we conducted ablation studies on the key components of MetaAgent: tool-using, traceback, and iteration.

**Tool-Using**  Tool-using is a crucial part of the finite state machine. When equipped with tools, the task-solving agent of a state can interact with the file system or the internet to solve complex tasks. The condition verifier will help to analyze the tool feedback as well, establishing a multi-turn interactive environment for tool-using, which can enhance the performance of the finite state machine. As the result in Table 5, the performance has decreased when the tool is disabled, showing that utilizing a search engine as a tool can help the agent clarify the answers and reach a higher score.

**Traceback**  The state traceback feature also contributes a lot when solving complex and unpredictable tasks. In the case that the current agent finds the input information needs to be refined via the previous state, the finite state machine enables traceback to the previous one and transmits the information to that agent. This design ensures the finite state machine is better at handling various situations, which distinguishes it from common linear structures like SOPs. The result of the ablation experiments also proves the assertion. In particular, we find that multi-agent systems without a traceback design often fail due to unresolved bugs. For instance, when the tester discovers a bug while executing the software code, they cannot relay this information back to the programmer without a traceback mechanism.

**Interation**  When designing the multi-agent system, a few iterations are required to make the system more robust. After testing the initial version of the multi-agent system on the pertinent test cases, the multi-agent system will be adapted in the aspect of agent and state design. The iteration can get rid of some unnecessary agents or intermediate states to simplify the work pipeline and enhance robustness. Results in Table 5 show that a sharp decrease in performance is caused by the absence of iteration. And in the bad cases, we do observe that the system struggles to complete the task due to excessively long text caused by unnecessary steps.

| Methods | ML_Bench | | Software | | Trivial Creative Writing | |
|---|---|---|---|---|---|---|
| | Score | $\Delta$(%) | Score | $\Delta$(%) | Score | $\Delta$(%) |
| MetaAgent (w/o tool-using) | – | – | – | – | 0.79 | ↓ 8.1 |
| MetaAgent (w/o iteration) | 0.61 | ↓ 26.5 | 0.65 | ↓ 35.3 | 0.65 | ↓ 24.4 |
| MetaAgent (w/o traceback) | 0.72 | ↓ 13.3 | 0.35 | ↓ 58.8 | 0.77 | ↓ 10.5 |
| MetaAgent | 0.83 | 0.00 | 0.85 | 0.0 | 0.86 | 0.00 |

Table 5: Comparison of Methods Across Different Tasks. ("–" means not applicable)

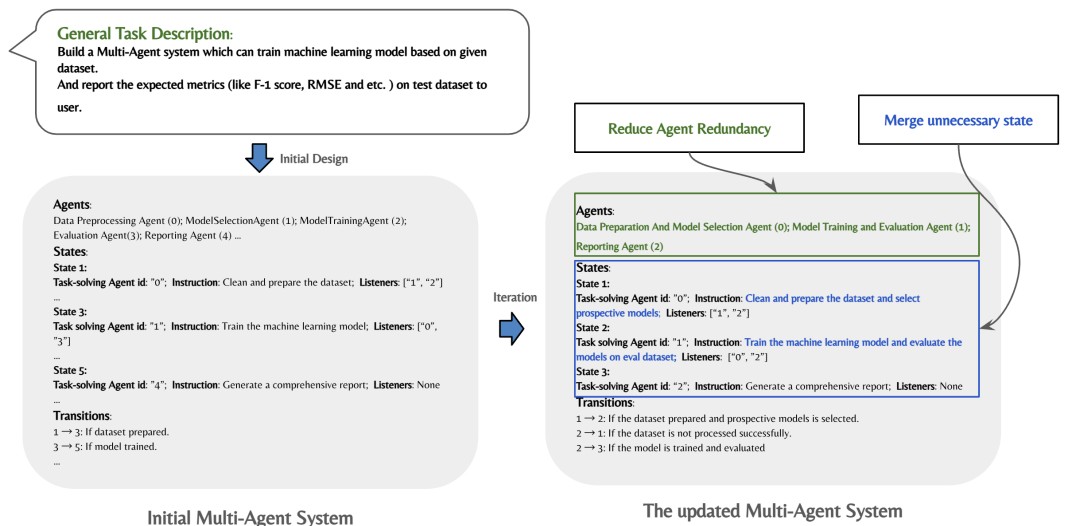

Figure 4: A Case Study Conduct on the Construction Stage

## 4.4 CASE STUDY

We present a case study comparing the initial multi-agent system with an updated version, using Machine Learning Bench as an example. Figure 4 illustrates the process of reducing agent redundancy and merging unnecessary states. Initially, the designer created a complex multi-agent system with five agents and five states. However, some agents had roles too trivial to justify their existence. For example, the "Evaluation Agent" could be merged with the "Model Training Agent," and the training and evaluation states could be combined. During the iteration process, we find the initial multi-agent system failed on generated tests due to overly long chains and trivial tasks. Due to the excessively frequent information transitions, agents experience a heavy burden on their memory, leading to the loss of important outputs to some degree. Additionally, because the states are too trivial, many agents have significant overlap in their tasks, which further reduces efficiency. After passing the trajectories to the adaptor, the system was updated and redundant agents and states were merged. The updated multi-agent system, with more integrated agents and states, performs much better than the initial version.

## 5 CONCLUSION

In this paper, we introduce MetaAgent, a framework that automatically generates multi-agent systems based on finite state machines. This approach addresses the drawbacks of both human-designed and auto-designed multi-agent systems. The finite state machine structure endows the generated multi-agent systems with tool-using and traceback capabilities. Additionally, the auto-design pipeline during the construction stage ensures that the multi-agent system is generally applicable to most cases within a task domain and can conduct self-iteration without external data. Experiments on practical tasks demonstrate the potential of MetaAgent. Automation is a growing trend in the LLM-based agent area, and MetaAgent provides a novel method for more practical scenarios.

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

## A    GENERAL TASK DESCRIPTIONS

**Software Development Task**    Build a multi-agent system that develops software. The multi-agent system could also save the developed software to a local file system and write a README for the user.

**Machine Learning Task**    Build a Multi-Agent system that can train a machine-learning model based on the given dataset. And report the expected metrics (like F-1 score, RMSE and etc. ) on the test dataset.

**Trivial Creative Writing Task**    Build a Multi-Agent System that can input a list of questions and then output a story that includes answers to all the questions in the list.

## B    SOFTWARE TASKS

**Evaluation Criteria**    We design several evaluation criteria for each software development task. Table 6 demonstrates on the criteria.

| Task Name | Evaluation Criteria |
|---|---|
| **2048game** | 1. Can open an interface
2. Can operate normally
3. Can merge correctly
4. Can score correctly |
| **Snake Game** | 1. Can open an interface
2. Can operate the snake normally
3. Can eat beans correctly
4. The snake can grow normally |
| **Brick Breaker Game** | 1. Can open an interface
2. Can operate the paddle normally
3. Can eliminate bricks correctly
4. Can score correctly |
| **excel app** | 1. Can open an interface
2. Can transfer files correctly
3. Can display correctly
4. Can close correctly |
| **weather** | 1. Can open an interface
2. Has weather query function
3. Can fetch weather data correctly
4. Can display weather data aesthetically |

Table 6: Evaluation Criteria for Software Development Tasks

## C    EXAMPLE MULTI-AGENT SYSTEMS

Here is an example Multi-Agent System for Software Development

```
{
```

```
756     "agents": [
757         {
758             "agent_id": "0",
759             "name": "RequirementDesigner",
760             "system_prompt": "You are RequirementDesigner. Your goal is
761                 to understand the software requirements and create a
762                 design or architecture for the software. Your
763                 responsibility is to gather and analyze the requirements
764                 for the software project and ensure that the design is
765                 robust and scalable.",
766             "tools": [
767                 "search_engine"
768             ]
769         },
770         {
771             "agent_id": "1",
772             "name": "CodeDeveloper",
773             "system_prompt": "You are CodeDeveloper. Your goal is to
774                 write the actual code for the software based on the
775                 design provided by RequirementDesigner. You are also
776                 responsible for writing a README file for the user and
777                 saving the developed software to a local file system.
778                 Ensure that the code is clean, efficient, and functional
779                 .",
780             "tools": [
781                 "file_writer"
782             ]
783         },
784         {
785             "agent_id": "2",
786             "name": "Tester",
787             "system_prompt": "You are Tester. Your goal is to test the
788                 software to ensure it works as intended. Your
789                 responsibility is to identify and report any bugs or
790                 issues in the software. You should also report the
791                 expected metrics on the test dataset to the user.",
792             "tools": [
793                 "code_interpreter"
794             ]
795         }
796     ],
797     "states": {
798         "states": [
799             {
800                 "state_id": "1",
801                 "agent_id": "0",
802                 "instruction": "Gather and analyze software requirements
803                     and create a design or architecture based on the
804                     requirements.",
805                 "is_initial": true,
806                 "is_final": false,
807                 "listener": [
808                     "1"
809                 ]
            },
            {
                "state_id": "2",
                "agent_id": "1",
                "instruction": "Write the actual code based on the design
                    , write a README file, and save the developed
                    software to a local file system.",
                "is_initial": false,
                "is_final": false,
                "listener": [
                    "2"
```

```
                    ]
                },
                {
                    "state_id": "3",
                    "agent_id": "2",
                    "instruction": "Test the software to ensure it works as
                        intended. Report the expected metrics (like F-1 score
                        , RMSE, etc.) on the test dataset to the user.",
                    "is_initial": false,
                    "is_final": false,
                    "listener": [
                        "0",
                        "1"
                    ]
                },
                {
                    "state_id": "4",
                    "agent_id": "0",
                    "instruction": "<|submit|> The a response to the user,
                        example: <|submit|>The software is developed and the
                        metrics on the test dataset are reported.",
                    "is_initial": false,
                    "is_final": true,
                    "listener": []
                }
            ],
            "transitions": [
                {
                    "from_state": "1",
                    "to_state": "2",
                    "condition": "If requirements are clear and complete and
                        design is robust and scalable"
                },
                {
                    "from_state": "2",
                    "to_state": "3",
                    "condition": "If code is clean, efficient, and functional
                        and README is clear, informative, and easy to
                        understand"
                },
                {
                    "from_state": "3",
                    "to_state": "4",
                    "condition": "If the software works as intended and
                        metrics are reported"
                },
                {
                    "from_state": "3",
                    "to_state": "2",
                    "condition": "If the test is not passed"
                }
            ]
        }
}
```

## D PROMPTS

### D.0.1 MULTI-AGENT SYSTEM GENERATION

```
You are the designer of a multi-agent system. Given a general task
    description and a list of agents, you need to generate a Finite State
    Machine (FSM) to manage the process of solving the task.
```

```
WARNING: You are good at controlling costs, too many agents and too
    complex cooperation structure can lead to excessive costs of
    information exchange
Each state in the FSM should include:
1. state_id: A unique identifier for the state
2. agent_id: The ID of the agent associated with this state
3. instruction: What the agent should do in this state
4. is_initial: Boolean indicating if this is the initial state
5. is_final: Boolean indicating if this is a final state
6. listener: The agent who will save this state output information in
    their memory
            Notice : Make sure the listener covers all related
                agents. The agents not listed as a listener would
                not received the information(which may cause the
                failure of cooperation)
            Hence, some important milestone like a new version of
                code/answer should be broadcast all related a g e n t

The FSM should also include transition functions between states. Each
    transition function should specify:
1. from_state: The ID of the state this transition is from
2. to_state: The ID of the state this transition goes to
3. condition: A description of the condition that triggers this
    transition

Your answer should follow this format:
Reasoning: <Your step-by-step reasoning process>
Answer:
```json
{{
  "states": [
    {{
      "state_id": "1",
      "agent_id": "0",
      "instruction": "Perform task X",
      "is_initial": true,
      "is_final": false,
      "listener":["1","2"]
    }},
    ...
  ],
  "transitions": [
    {{
      "from_state": "1",
      "to_state": "2",
      "condition": "If task X is completed successfully"
    }},
    {{
      "from_state": "2",
      "to_state": "1",
      "condition": "If the previous task needs to be re-done."
    }},
    ...
  ]
}}
```

Rules:
1. Ensure there is exactly one initial state and at least one final
    state.
2. Every non-final state should have at least one outgoing transition
    .
3. The FSM should be able to handle loops and complex interactions
    between agents.
```

```
        4. Include a transition to a final state that submits the final
            answer (use <|submit|> in the instruction).
        5. Make sure all agent_ids in the states correspond to the provided
            agent_dict.
        6. The transitions should consider as many as possible situations.
            Which consisit a roadmap for Multi-Agent System in deployment
            stage.
```

### D.0.2 UPDATING THE MULTI-AGENT SYSTEM

```
You are a Multi-Agent System Designer. Your task is to modify the Multi-
    Agent System based on the existing failed task cases.

The goal that this Multi-Agent System needs to solve is: {
    task_description}

The current structure of the Multi-Agent System is as follows:

Part 1: Agent Design:

Each agent contains three features:

1. name: <The name of the agent>
2. system_prompt: <The system prompt for the agent, describing the
    overall goal, its name and role, and its responsibility and
    constraints.>
3. tools: <The equipped tool name, a list>
Part 2: Communication System Design:

We use a finite state machine (FSM) to manage the cooperation of agents.
    Specifically:

Each state in the FSM should include:

1. state_id: A unique identifier for the state
2. agent_id: The ID of the agent associated with this state
3. instruction: What the agent should do in this state
4. is_initial: Boolean indicating if this is the initial state
5. is_final: Boolean indicating if this is a final state
6. listener: The agent who will save this state's output information in
    their memory
Notice: Make sure the listener covers all related agents. The agents not
    listed as a listener would not receive the information (which may
    cause the failure of cooperation). Hence, some important milestones
    like a new version of code/answer should be broadcast to all related
    agents!
The FSM should also include transition functions between states. Each
    transition function should specify:

1. from_state: The ID of the state this transition is from
2. to_state: The ID of the state this transition goes to
3. condition: A description of the condition that triggers this
    transition
Both parts are represented in JSON, forming a Multi-Agent System.

The current goal for the Multi-Agent System is: \n {task_description}

The existing Multi-Agent System is: \n {MAS}

While using this Multi-Agent System to solve the problem, it failed: \n {
    bad_cases}

Please think step by step to optimize the existing Multi-Agent System.
Gradually output your thought process.
```

```
WARNING: The number of agents and the number of states should be
    minimized as much as possible. For saving the token cost!

What are the specific reasons for the failure in the above bad cases?
    What aspects were not considered, and how can we improve them from
    the following aspects?
Is the current role positioning of the agents reasonable? Are these
    agents necessary to solve this task, or do we need to create new
    agents? (DO NOT ADD AGENT UNLESS IT IS NECESSARY)
Is the current communication structure optimized to reduce the cost of
    information exchange? (DO NOT ADD STATE UNLESS IT IS NECESSARY)
Are the instructions for each state specific and feasible, and how can
    they be optimized?
Use add examples in the prompts to optimize the multi-agent system!
Now, output your thought process and output the new Multi-Agent System
    design in JSON format.

Please consider: 1. Whether the functionalities of multiple Agents can be
    integrated into a single Agent to reduce unnecessary communication
    exchanges. For example, Reasoning and Action should be placed within
    the same Agent. Please note that the essence of multi-Agent systems
    is to provide diverse perspectives, not to split task processes and
    forcibly create Agents. States should also be streamlined as much as
    possible; one state can accomplish many specific actions, rather than
    just one action. HOWEVER, THERE MUST BE A FINAL STATE SPECIALLY FOR
    SUBMITTION , where the agent shold use <|submit|> to submit the final
    answer. Beacuse when the states transfer to final state, the finite
    states machine will be shut down. So the final states should contain
    and only contain the 'sbumit'
2. Why the tasks failed? Can the Agent Description or Tool assemble can
    be updated?
3. How to optimize the performance? Modify the FSM or the instruction of
    each state? (eg. Try and compare different ML models )
```json
<fill in your Multi-Agent System Design (Agents and FSM)>
```
```