# OpenReview forum: "MetaAgent: Automatically Building Multi-Agent System based on Finite State Machine"
_ICLR.cc/2025/Conference — Submitted to ICLR 2025_

### Official Review · Reviewer_wa4C · 2024-10-21

**Soundness:** 2
**Presentation:** 2
**Contribution:** 2
**Rating:** 3
**Confidence:** 4

**Summary:**

This paper introduces the Meta Agent, automatically building multi-agent systems based on finite state machines. The authors validate the advantages of the Meta Agent through testing on three benchmarks.

**Strengths:**

1. Applying finite state machines to the Agentic Workflow is an interesting and intuitive idea.

2. The author's experiments cover a wide range, including machine learning bench, software development, and NLP tasks (Trivial Creative Writing). These settings serve as excellent benchmarks for testing agent systems. Additionally, the author provides extensive experiments across three settings to strengthen the experimental evidence.

**Weaknesses:**

1. The motivation behind this paper is not clear. The author attempts to address issues like the lack of tool support, external data integration, and inflexible communication structures. However, the proposed meta-agent based on a finite state machine (FSM) does not clearly convey why FSM is suitable for solving these problems.

    In the implementation process, the meta-agent involves various components such as tool-using and self-iteration that seem unrelated to FSM. Despite the experimental results showing that the meta-agent achieves state-of-the-art (SOTA) performance on multiple tasks, the agent system feels ad-hoc and trivial, which significantly reduces the novelty and technical depth of the paper.

    The ablation study in section 4.3 confirms this point—when any of the three modules is removed, the model's performance falls short of the claimed level, further reinforcing concerns that the framework may be over-desined and ad-hoc.

    This issue seems to be a common problem in agent research, where a fancy key idea is proposed, but the implementation relies on combining various components to achieve performance gains. This approach to innovation greatly hinders deeper development in the field of agents. The author could focus more deeply on the advantages of FSM, such as trace-back analysis, providing not only empirical evidence but also theoretical insights into why an agent based on such a structure could be further improved. This would enhance the depth of the paper.

2. The content of the paper, including the appendix, lacks specific descriptions of the implementation. For instance, in Figure 2, how each core module of the meta-agent is implemented—whether it relies on prompting or workflow—is unclear. Similarly, in the software development task, the author mentions "earning one point for each test it passes," but does not specify how it is determined whether a test is passed or not. The author also not specify the version of GPT4o in experiment. These ambiguities make it difficult to reproduce the author's experiments.

3. The writing of the paper could be further improved. For example, Figure 1 is difficult to follow. It involves an agentic workflow with many details, such as using search tools, feedback mechanisms, and state traceback, which in turn involve various agents like “product manager” and “information collector” that are not explained in the main text. The author should use illustrations to help readers understand the process clearly. Perhaps combining Figure 1 with the content in section 3.2 could better explain which stages each step in the figure corresponds to. Additionally, Figure 2 lacks a clear breakdown of the information it presents, making it difficult for readers to understand.

4. Since the meta-agent involves stages like Agent Design, Self-Iteration, and condition verification, it inevitably introduces more inference costs. The author lacks a systematic discussion and analysis of the inference cost associated with the proposed framework.

5. The prompts in Appendix D appear to be very complex and specific, raising concerns about whether the proposed method relies too much on prompt design and whether the method is stable with minor changes to the prompts.

**Questions:**

Please refer to my comments in Weaknesses. In addtion to these comments, one more question is:

1. Line 453, why tool-using is a crucial part of FSM?

---

### Official Review · Reviewer_U996 · 2024-10-30

**Soundness:** 2
**Presentation:** 1
**Contribution:** 2
**Rating:** 3
**Confidence:** 4

**Summary:**

The paper proposes MetaAgent, a method to automatically construct a multi-agent system. It uses finite-state machines to manage the communication between different agents within the system. The method is evaluated on ML-Benchmark, some software engineering tasks, and Trivial Creative Writing.

**Strengths:**

1. The paper is well-motivated to automate the process of multi-agent system design and use finite-state machines for non-linear problem-solving procedures. Automatic multi-agent system design is also a timely research topic.
2. Although they lack sufficient discussions and details in the paper, the test case generation and self-refinement stages in the proposed method can be interesting.

**Weaknesses:**

1. Lack of technical details about the proposed method and discussions/analysis to support the method .
- The only technical details that can be inferred from the Appendix is that the initial finite-state machine (FSM) generation and iterative refinement is implemented by prompting a certain LLM. Yet, the authors did not mention which LLM is used and what are the settings (e.g. in-context examples, generation hyperparameters, etc.).
- There should be an intrinsic evaluation of automatically generated FSMs by comparing them with expert-designed FSMs to support the method and analyze its errors for better understanding.
- The remaining parts of the proposed method, agent design and test case generation, have limited descriptions of their implementation.
- Is the design of each agent also generated by a certain LLM? If so, what is the prompting/training strategy here?
- How are the test cases generated and executed on the FSM? How do you automatically come up with corner cases?
- The benefit of FSM over non-deterministic approaches to support non-linear workflows and allow backtracking is unclear. For example, AutoGen uses an LLM-based chat manager to decide which agent to call next and pass outputs from different agents around. Such non-deterministic designs are more flexible in error-handling, backtracking, and dynamic workflows.

2. Lack of justification and rationales in some experimental design.
- It is unclear why some baseline methods cited in the paper, such as EvoAgent, are not included in the experiments for machine learning and software engineering tasks.
- Six software engineering tasks are proposed as part of the evaluation. However, their source, data collection, formulation, etc. are not discussed in the paper, making it less credible. Additionally, the evaluation criteria of these six tasks mentioned in Appendix B largely simplifies the real-world setting and only considers some naive conditions, which makes the comparison among methods questionable.

**Questions:**

The paper would benefit from another round of revision to fix the following issues with automatic writing checkers and AI writing assistants:
1. The in-text citation style is not correct throughout the paper. Please use `\citep` instead of `\citet` if not using the citation as a subject of a sentence. Please also remove redundant parentheses and use commas instead of semicolons to separate citations.
2. There are quite a few typos and grammar issues in the manuscript, some of which are impeding the understanding of this paper. For instances:
- “, the results” -> “. The results” (line 21)
- “complement various complex tasks” -> “complete various complex tasks”? (line 31)
- “has proposed” -> “has been proposed” (line 34)
- “relative works” -> “related work” (line 107)
- “Interation by itself” what is “interation”? (line 299)
- “We the initial version” -> “The initial version”? (line 300)
3. As specified in ICLR 2025 submission template, table captions should be placed at the top, not bottom.

---

### Official Review · Reviewer_ZFfR · 2024-11-03

**Soundness:** 3
**Presentation:** 3
**Contribution:** 3
**Rating:** 6
**Confidence:** 4

**Summary:**

This paper proposes a framework for designing multi-agent systems based on finite state machine, MetaAgent. MetaAgent can automatically generate a system according to the task description, including decomposing the whole task, assigning the role of each agent, and equipping them with tools. The agents of the system execute sequentially, and it supports state traceback when encountering mistakes. After initializing the system, this framework can iteratively refine the system through self-generated queries. The results of experiments show that MetaAgent surpasses other baselines in both real-world coding tasks and NLP tasks.

**Strengths:**

1. The logic of this paper is clear. On the one hand, it illustrates the components of MetaAgent and makes it understandable. On the other hand, the paper shows the difference between MetaAgent and others, clearly pointing out the advantages of MetaAgent.
2. This paper introduces finite state machine for designing multi-agent systems. It enables state traceback and self-refine, enhancing the robustness of the system.
3. MetaAgent does not need external training data and can iteratively refine without carefully handcrafting test data.

**Weaknesses:**

1. This method heavily depends on the capability of the base model, especially the designer model, as the designer needs to decompose the whole task to an appropriate level for each agent and assign tasks to them.
2. There are some typos:  missing '(', ')' in line 39,40;  'We' should be deleted in line 300.

**Questions:**

1. In section 3.2, how is an edge case created in test case generation? In other words, how is 'edge' defined given a task description?
2. In section 3.2, how do states refine when encountering mistakes? Do they change instructions or tools?
3. What is the fundamental reason why MateAgent performs well on some difficult tasks? For example, in ML bench, MateAgent is significantly better than AutoAgents with a result of 0.
4. How can MateAgent be applied to other questions within the same domain? Is it necessary to refine the system using all cases in this task domain?

---

### Official Review · Reviewer_BEdi · 2024-11-04

**Soundness:** 2
**Presentation:** 2
**Contribution:** 2
**Rating:** 5
**Confidence:** 3

**Summary:**

The paper introduces MetaAgent, a framework that autonomously generates multi-agent systems with a finite state machine (FSM). This system aims to address limitations in current human-designed and auto-designed frameworks, such as adaptability to specific scenarios, reliance on external data, and lack of robust communication structures.

The MetaAgent framework operates by designing agents and configuring them within an FSM. The system starts by generating a set of agents, including names, system prompts, and tools, based on the task's general requirements. It then establishes multiple states representing different task-solving stages, each linked to an agent with specific instructions and conditions for state transitions. Each state includes a condition verifier that monitors agent outputs to determine if transitions or refinements are needed and listeners that receive the output. After the initial design, MetaAgent conducts an iteration phase. It uses generated test queries to identify and resolve system deficiencies, optimizing the agent's design, FSM structure, and prompts.

The authors validate MetaAgent's efficacy through experiments on two coding tasks: machine learning benchmark and software development, and an NLP task: trivial creative writing. The results demonstrate that MetaAgent performs better than existing auto-designed MAS frameworks and matches or surpasses human-designed MAS tailored for specific tasks. Ablation studies further underscore the critical roles of MetaAgent's features, like tool usage, iteration, and state traceback, in achieving these results.

**Strengths:**

1. The paper addresses a crucial problem in automating the design of LLM agents without heavy human-coded task-specific implementation. This automation is key to developing agent systems that can generalize across diverse tasks and evolve independently, which could significantly expand the applicability of LLM-based multi-agent frameworks.

2. The proposed framework effectively combines multiple elements to construct a multi-agent system, including an initial design of agents, a finite state machine (FSM) for effective task execution, and self-iteration through model-generated test queries. This design is both innovative and comprehensive, making the system adaptable and robust.

3. Preliminary experimental results validate the effectiveness of MetaAgent, showing strong performance across several tasks and competitive results compared to both human-designed and existing auto-designed systems.

**Weaknesses:**

1. The main weakness I am concerned about is the limited experimental scope. The experiments include only two coding tasks and one NLP task. The coding tasks themselves may be subject to data leakage, as they rely on common ML datasets and applications likely encountered by LLMs in their pretraining. A key goal of automating LLM agent design is to demonstrate true generalization across diverse, practical tasks. To strengthen the evaluation, testing on broader benchmarks like AgentBench for task variety and SWE-bench for challenging software engineering (SWE) tasks would provide more robust evidence of generalization.

2. Simplistic agent design in case study and experiment. The case study's agent design is relatively simple, involving only two or three states. This raises concerns about the framework’s scalability to more complex tasks requiring intricate agent designs. For instance, general SWE tasks involving real-world code repositories may demand a more elaborate FSM structure, which the paper currently does not address. Demonstrating scalability to these more complex scenarios would enhance the method’s practical relevance and applicability.

**Questions:**

1. I would suggest testing on a wide range of tasks. E.g., use AgentBench (1).
2. In all experiments, include an agentless baseline, i.e., a human-designed hard-coded procedural LLM task solver, with the same tool use. Similar to the data interpreter baseline in the ML-bench experiment, it should also be added to the other experiments (My understanding is that the COT baseline for the NLP task does not use tools ?).
3. Add some stats/descriptions on the agent system/state machine generated for each task. Including the overall design, such as the number of agents and states. As well as runtime stats like how often the agent backtrace, self-iterate etc. Which would be helpful for understanding the operation of the generated agents.


1. Liu, Xiao, Hao Yu, Hanchen Zhang, Yifan Xu, Xuanyu Lei, Hanyu Lai, Yu Gu et al. "Agentbench: Evaluating llms as agents." arXiv preprint arXiv:2308.03688 (2023).

---

### Meta-Review · Area_Chair_CcPZ · 2024-12-23

**Metareview:**

The paper introduces MetaAgent, a framework for automatically constructing multi-agent systems using finite state machines (FSM). While it addresses an important problem, the paper has several critical weaknesses. The experimental scope is narrow, focusing on two coding tasks and one NLP task, limiting the evaluation of its generalization capabilities. Key methodological details, such as FSM implementation, test case generation, and agent design, are poorly explained, impeding reproducibility. The paper also lacks sufficient theoretical justification for using FSMs over alternative methods and fails to analyze the practical trade-offs, such as computational costs. Additionally, the writing quality, including unclear figures and inconsistencies, detracts from the paper’s overall clarity.

Strengths include addressing automation in multi-agent system design and the introduction of self-iteration and state traceback. However, these are overshadowed by the lack of detailed implementation, narrow evaluation, and limited novelty.

**Additional Comments On Reviewer Discussion:**

The rebuttal clarified some aspects, such as the FSM's role and examples of agent design, but did not sufficiently address concerns about the limited evaluation, unclear implementation details, and lack of theoretical grounding. Reviewers remained unconvinced of the framework's scalability and practical relevance. Despite efforts to provide additional explanations, the methodological and experimental limitations justify the rejection, with encouragement for substantial revisions.

---

### Decision · Program_Chairs · 2025-01-22

Reject